# Characterization, Antioxidant and Anti-Inflammation Capacities of Fermented *Flammulina velutipes* Polyphenols

**DOI:** 10.3390/molecules26206205

**Published:** 2021-10-14

**Authors:** Sheng Ma, Hongcai Zhang, Jianxiong Xu

**Affiliations:** 1School of Agriculture and Biology, Shanghai Jiao Tong University, Shanghai 200436, China; 1329109669@sjtu.edu.cn; 2Shanghai Key Laboratory for Veterinary and Biotechnology, Shanghai 200436, China

**Keywords:** *Flammulina velutipes* polyphenols, fermentation, anti-inflammation, antioxidant capacities, NLRP3 signal pathway

## Abstract

This work investigated the preparation, characterization, antioxidant, and anti-inflammation capacities of *Flammulina velutipes* polyphenols (FVP) and fermented FVP (FFVP). The results revealed that the new syringic acid, accounting for 22.22%, was obtained after fermentation (FFVP). FFVP exhibits higher antioxidant and anti-inflammation activities than FVP, enhancing cell viability and phagocytosis, inhibiting the secretion of NO and ROS, and reducing the inflammatory response of RAW264.7 cells. This study revealed that FFVP provides a theoretical reference for in-depth study of its regulatory mechanisms and further development of functional antioxidants that are applicable in the food and health industry.

## 1. Introduction

*Flammulina velutipes* (FV), belonging to the family Basidiomycetes, has exceptional antioxidant, anti-tumor, and cholesterol-reducing abilities due to its multiple functional compounds, including polyphenols, polysaccharides, glycoprotein, and fungal immunomodulatory protein (FIP) [1,2]. Among these, polyphenols, derived from the metabolic pathway of phenyl propane, contain multiple hydroxyl groups and at least two phenolic rings and could eliminate excessive free radicals by joining the reactive oxygen species and active nitrogen [3,4]. Furthermore, FV also alleviates human skin cell damage caused by different oxidants, reduces ROS concentrations, lipid peroxidation and DNA damage, and improves mitochondrial function [5]. The extraction and functional activities of polyphenols obtained from plants and vegetables have been reported in previous studies [6,7]. However, very few studies have focused on the enhanced biological capacities of polyphenols via microbial fermentation.

It is a known fact that microbial fermentation enriches the biological compounds and antioxidant capacities of fermentation products [8]. Shumoy et al. (2017) reported that the contents of free and bound polyphenols improved significantly with an extended fermentation time [9]. The better antioxidant abilities of fermentation polyphenols alleviate oxidative stress injury caused by the imbalance of ROS/RNS, radiation-induced free radicals, and contact with heavy metals [10,11,12]. Wang et al. (2013) showed that oxidative stress was highly associated with chronic and acute diseases, including cancer, neurological disorders, cardiovascular disease, diabetes, etc. [13].

To activate the NLRP3 inflammasome, the NF-κB pathway is used as a priming signal [14]. NF-κB p65 is usually bound to the cytoplasm in an inactive form with the IKB inhibitory factor. When bacteria, viruses, or cytokines stimulate the cells, IKB and NF-κB p65 are degraded by activated IKK; thus, the latter is activated in the nucleus to perform its functions [15]. One of the most important antioxidant signaling pathways in the body is the NLRP3 signaling pathway [16]. It is triggered by lipopolysaccharide (LPS) and mediated by the host’s immune response to microbial infection and cell injury via accumulation of the NLRP3 inflammasome. This further results in cleavage of pro-caspase-1 protein to produce activated Caspase-1, and the transformation of cytokine precursors Pro-interleukin-1β (Pro-IL-1β) and Pro-interleukin-18 (Pro-IL-18) into mature IL-1β and IL-18 [17].

There has been no study of the isolation and biological capacities of *Flammulina velutipes* polyphenols (FVP) and fermented FVP (FFVP) to date. We theorized that FFVP could enhance the antioxidant and anti-inflammation abilities of polyphenols via the NLRP3 signaling pathway. Therefore, the main objective of the present study was to compare the antioxidant capacities, mRNA, and expression levels of proteins against LPS-induced oxidant stress between FVP and FFVP. Furthermore, the anti-inflammation mechanisms of FVP and FFVP were further highlighted in the RAW264.7 cells.

## 2. Results

### 2.1. Polyphenol Yield and Purity

The results exhibited that the polyphenol yield of FVP and FFVP was 0.89% and 0.98%, respectively. Fermentation amplified the polyphenol yield of *Flammulina velutipes* by 10.11%. The purity of the purified polyphenols was 85.29% and 85.74%, respectively.

### 2.2. HPLC and GPC Analysis

FVP and FFVP had similar amounts of gallic acid and ferulic acid (Figure 1 and Table 1). Compared with FVP, rutin contents in the FFVP were significantly decreased, but quercetin contents reached 30.30% (FVP, 15.56%). Moreover, the phenolic compounds and proportions of FFVP were changed. A new syringic acid accounting for 22.22% was determined in FFVP but not detected in FVP. The possible reason for the increased phenolic contents via fermentation was the synthesis of new phenolic compounds due to the fugal metabolism and bioconversion of metabolites into soluble forms [18]. Schmidt et al. (2014) showed that chlorogenic acids, p-hydroxybenzoic acids, and vanillin contents in fermented rice bran were amplified after fermentation [19]. In contrast, gallic acid and caffeic acid increased at 72 h, and syringic and ferulic acids increased after fermentation at 120 h. These results are consistent with the present study.

### 2.3. Morphology Analysis of FVP and FFVP

FVP has an irregular shape, rough surface, and cavity structure (Figure 2A), whereas FFVP has a plate-like shape with a relatively flat surface (Figure 2B). The morphology results of FVP and FFVP were in line with the previous report [20], in which β-pyran polysaccharides from bamboo shoot shells had a sheet-like shape, indicating strong interaction forces between polysaccharides chains, due to its highly branched chain conformation. FFVP had several branches, a good rehydration performance, and a strong aggregation bond, indicating a strong interaction between the highly branched polyphenol chains [21].

### 2.4. Antioxidant Capacities of FVP and FFVP

The antioxidant capacities of FVP and FFVP are shown in Figure 3. The antioxidant powers of polyphenols were concentration-dependent. In general, the antioxidant capacities of FFVP were higher than that of FVP at the same concentration. The DPPH radical scavenging rate of FFVP at 400 and 500 mg/L, OH· scavenging rate at 800 mg/L, and total reducing power at 300 and 400 mg/L were significantly higher than that of FVP (*p* < 0.05). The absolute reducing power of FFVP (0.38) at 500 mg/L was greater than that of FVP (0.24). The previous study reported that fermentation could improve the bioavailability of polyphenols in vivo since the hydrolases produced during fermentation releases abundant polyphenols from plant raw materials, thus increasing the free content of polyphenols and their antioxidant capacities [22]. Shin et al. (2018) also reported that polyphenol contents and free radical scavenging capacities of the ethanol extract from black rice bran increased drastically after *Aspergillus oryzae* fermentation [23]. Moreover, Sun et al. (2014) also said that the antioxidant capacities of polyphenols were related to the conformation and molecular structure of polyphenols, indicating that lactic acid bacteria fermentation could change the yield and structure of polyphenol metabolites [24,25]. Hence, the changes in polyphenol contents and ratios of FFVP contributed to their enhanced antioxidant capacities.

### 2.5. Anti-Inflammation Capacities of FVP and FFVP on RAW264.7 Cells

#### 2.5.1. Cell Viability

The effect of LPS concentration on the viability of RAW264.7 cells is revealed in Figure 4A. The cell viability of RAW264.7 reduced with increasing concentrations of LPS and reached 39.75% at 1 µg/mL for 24 h. Wang et al. (2013) reported that LPS causes inflammation in macrophages at 1 µg/mL, which was consistent with the present study [4]. Thus, an optimal concentration of 1 µg/mL was chosen for further research.

The effect of polyphenol concentration on cell viability is shown in Figure 4B. The cell viability of the FFVP group was higher than that of the FVP group at all concentrations but showed a significant difference at 100 µg/mL (*p* < 0.01). When polyphenol concentration was less than 100 µg/mL, the cell viability was enhanced with increasing polyphenol concentration. However, cell viability was reduced when the concentration was higher than 100 µg/mL. Therefore, 25, 50, and 100 µg/mL polyphenols were carefully chosen for further studies.

#### 2.5.2. NO, ROS, and Phagocytosis Analysis

FVP and FFVP’s effect on the secretion of NO and ROS in RAW264.7 cells is shown in Figure 5. Compared with the CON group, the NO and ROS contents in the LPS group were significantly increased (*p* < 0.01) with polyphenols. The NO and ROS contents in the FFVP group were lower than in the FVP group at the same concentration (*p* < 0.05). Additionally, NO and ROS contents in the FFVP group were significantly lower than that of the FVP at 100 µg/mL (*p* < 0.01). The previous studies confirmed that polyphenols with antioxidant capacities could effectively clear the harmful ROS and inhibit antioxidant damage by reducing ROS secretion [26]. The effect of FVP and FFVP on phagocytosis in RAW264.7 cells is exhibited in Figure 6. Compared with the LPS group, the phagocytic function of RAW264.7 cells was enhanced after adding polyphenols, and that of the FFVP group was higher than that of the FVP group at the same concentration. Cristina’s study indicated that polyphenols could enhance splenic NK cell-killing activity and the phagocytosis of peritoneal macrophages by improving the phagocytosis rate, expressed as phagocytosis frequency, (i.e., percentage of phagocytosing cells/total cells) [27]. The mean fluorescence intensity (MFI) of FFVP reached 14.86 at 100 µg/mL and was significantly higher than the others.

#### 2.5.3. Secretion and Expression of Inflammatory Cytokines

Compared to the control group, the contents (Figure 7) and expression (Figure 8) of inflammatory cytokines (IL-1β, IL-6, IL-18, and TNF-α) in the LPS group were significantly higher (*p* < 0.05), and the inflammatory cytokines in the supernatants of FVP and FFVP reduced with increasing polyphenols concentration. Furthermore, the contents and expression of inflammatory cytokines in the FFVP group were lower than that in the FVP group at the same concentration. Shakoor et al. also reported that the polyphenols could reduce inflammation by suppressing the pro-inflammatory cytokines in inflammatory bowel disease [28]. In this study, FFVP aggregation was enhanced, and the monophenol species and proportion of polyphenols were changed after fermentation, improving the immunomodulatory effects of FVP [29].

#### 2.5.4. LPS-Induced NF-κB and NLRP3 Signal Pathway Inactivation Treated by FVP and FFVP

The mRNA expression of the NF-κB and NLRP3 signal pathways is shown in Figure 9. Induced LPS significantly activated the NF-κB and NLRP3 signaling pathways than the CON group (*p* < 0.01), but FVP and FFVP inhibited the signaling way. Additionally, FVP and FFVP reduced the phosphorylation and mRNA expression of NF-κB p65 and IkBα when compared to LPS treated alone. The FVP and FFVP’s effect on protein expression of NF-κB and NLRP3 at 25, 50, and 100 µg/mL was examined (Figure 10), revealing a similar mRNA expression. In addition, the relative NLRP3 and Caspase-1 level of the FVP group was higher than that of the FFVP group (*p* < 0.05). The possible mechanism was FFVP reducing the activation of the NLRP3 inflammasome by reducing ROS secretion and preventing the activation of the NF-κB signaling pathway.

## 3. Discussion

In this study, the antioxidant and anti-inflammation capacities of FFVP were found to be better than FVP. Compared with FVP, the quercetin contents reached 30.30% in the FFVP (FVP, 15.56%). Moreover, a new syringic acid accounting for 22.22% was determined in FFVP but not detected in FVP. FFVP had several branches, good rehydration performance, and a strong aggregation bond, indicating a strong interaction between the highly branched polyphenol chains.

The detailed molecular mechanisms leading to the activation of the NLRP3 signal pathway are still uncertain at present. Various molecular mechanisms have been suggested to explain the activation of the NLRP3 inflammasome, including ROS generation, intracellular ionic fluxes, and lysosomal destabilization [30,31]. For instance, Harijith et al. reported that ROS, highly reactive molecules mainly produced by reducing oxygen free radicals during mitochondrial oxidative phosphorylation, were considered a common trigger for NLRP3 inflammasome activation [32]. Furthermore, the most important proteins for the assembly, formation, and activation of NLRP3 inflammasome are TXNIP and MAVS proteins regulated by ROS [33]. ROS also provides a priming signal for NLRP3 inflammasome activation by activating the NF-κB signaling pathway, increasing NLRP3 and proIL-1β expression [34].

Many studies have shown that NF-κB regulates the NLRP3 signaling pathway [35,36]. Yu et al. reported that 4′-Methoxyresveratrol reduces the expression of NLRP3 and Caspase-1 by preventing the activation of NF-κB and decreasing the secretion of mature IL-1β [37]. Doss et al. indicated that ferulic acid (30 mg/kg)-treated rats reduced the mRNA expression of NF-κB p65 [38]. Thus, the protein expression of NLRP3, Caspase-1, and pro-inflammatory cytokines was found to be significantly low. In this study, both the FVP and the FFVP regulates the NLRP3 signaling pathway by inhibiting the expression of NF-κB.

Microorganisms degrade the cell-wall structure of FV and change the composition of polyphenols. The mechanism of polyphenol transformation should be highlighted by an in-depth analysis of bioconversion pathways such as glycosylation, deglycosylation, ring cleavage, methylation, glucuronidation, and sulfate conjugation [39]. The other metabolites of FV, including polysaccharides, flavones, phenolic acid and its microbial richness, should be extracted and investigated in further studies. Furthermore, the detailed molecular mechanisms of activation of NF-κB and NLRP3 signal pathway induced by FFVP were studied using advanced techniques to explain the regulation of polyphenols in vivo.

## 4. Materials and Methods

### 4.1. Materials and Reagents

FV was obtained by Shanghai Guangming Senyuan Biotechnology Co., Ltd., China. Chuangbo Microbial Starter including CMCCB 63501 (*Bacillus subtilis*), ATCC 15707 (*Bifidobacterium longum*), ATCC 9763 (*Saccharomyces cerevisiae* Meyen ex E.C. Hansen) was a kind gift from Shanghai Chuangbo Ecological Engineering Co., Ltd, China (production batch No. CB08190529). RAW264.7 (mouse mononuclear macrophage leukemic primary cells) was purchased from the Type Culture Collection of Chinese Academy of Sciences, Shanghai, China. Other chemicals and reagents were of analytical grade and purchased from Sigma-Aldrich (St. Louis, MO, USA).

### 4.2. Extraction and Purification of Polyphenols from FV

FV was pulverized with the Waring blender (Shanghai Shibang Machinery Co., Ltd., China) and passed through a 0.75 mm sieve. Chuangbo microorganism starter was then added to the smashed FV at 0.10%. The fermentation conditions, including the molasses contents (3%), temperature (28 °C), moisture contents (40%), and culture time (10 days), were optimized for FFVP production.

Supercritical CO_2_ fluid extraction (model HA131–50–01, Nantong supercritical extraction Co., Jiangsu, China) from FV and FFV, respectively, were used to extract the FVP and FFVP [40]. The optimized extraction conditions included an extraction time of 2 h, extraction temperature of 50 °C, extraction pressure of 30 MPa, extractor (ethanol) volume fraction of 60%, extraction sample amount of 4 BV and CO_2_ flow rate of 15 L/h. The crude polyphenols were further cleansed by column chromatography using macroporous adsorption resin as filler [41]. The conditions used had a pH of 4.0, 3 mg/mL concentration, loading volume of 4.5 BV, and ethanol concentration of 60%.

### 4.3. Morphology Analysis of FVP and FFVP

#### 4.3.1. High-Performance Liquid Chromatography (HPLC) Analysis

HPLC was used to analyze the polyphenol contents with a chromatographic column (ZORBAX SB-C18 column, Agilent, USA) and variable wavelength detector (VWD) (LC1200, Agilent, Palo Alto, USA) [42]. The mobile phases used were acetonitrile (A) and 0.4% acetic acid (B). The gradient elution programs of the mobile phase were as follows: 0–40 min, 5% A and 95% B; 40–45 min, 25% A and 75% B; 45–50 min, 35% A and 65% B. The operation time and flow time were set at 5 min and 1.0 mL/min, respectively. The discovery wavelength and column temperatures were 280 nm and 30 °C, respectively.

#### 4.3.2. SEM Analysis

The superficial characteristics of samples at 1000×, 5000×, and 10,000× magnifications were explained using SEM (FEI SIRION-200, Hillsborough, USA). One mg of FVP and FFVP was fixed on a sample holder, dried by a critical point dryer (LADD 28000, Williston, USA), and coated with a thin gold layer of 3 mm by a sputter coater (JBS E5150, Austin, USA) for conductivity [43]. The morphologies of samples were observed at an accelerating potential of 15 kV.

### 4.4. Antioxidant Capacities Analysis

DPPH radical scavenging rates of FVP and FFVP were determined according to previous reports [44]. The sample solution of 0.65 mL was mixed with 0.195 mL DPPH solution (0.1 mmol/L) and incubated for 30 min in the dark; the absorbance was determined at 517 nm. The scavenging capacity that corresponds to the percentage of DPPH radical scavenging was determined according to previous reports [44].

Hydroxyl radical scavenging rate was measured according to the previously reported method with minor revisions [45]. A total of 0.15 mL of 2-deoxyribose (5 mmol/L), 0.4 mL of sodium phosphate buffer (0.75 mol/L), 0.25 mL of distilled water, and 0.1 mL of ferrous sulfate solution (7.5 mmol/L) was mixed with 0.1 mL of the sample solution. The mixed solutions were incubated at 37 °C for 1 h, and the absorbance of samples was measured at 536 nm.

Superoxide anion radical scavenging rate was determined based on a previous report with slight modifications [46]. A total of 1 mL of nitrotetrazolazolium chloride solution (150 µmol/L), 1 mL of nicotinamide adenine dinucleotide solution (468 µmol/L), and 1 mL of phenazine methyl sulfate solution (60 µmol/L) were added to 1 mL of sample solution. The reaction was carried out after mixing for 5 min, and the absorbance was recorded at 560 nm.

Overall reducing power was determined according to the method of Yao et al. [47]. The acetate buffer solution (0.3 mol/L), 2,4,6-tripyridinyl triazine solution (10 mmol/L), and ferric chloride solution (20 mmol/L) were carefully mixed (10:1:1, *v*/*v*/*v*) and preheated at 37 °C. The distilled water (30 µL) and sample solution (10 µL) was then added. The total reducing power of FVP and FFVP was recorded at 593 nm at 37 °C for 30 min.

### 4.5. Anti-Inflammation Capacities of FVP and FFVP on RAW264.7 Cells

#### 4.5.1. Cell Viability Analysis

RAW264.7 cells were cultured for 12 h in a working solution containing 10% fetal bovine serum, 1% double antibody (100 IU/mL penicillin and 100 IU/mL streptomycin), and 90% Dulbecco’s modified eagle medium (DMEM) in an incubator (Heracell 240i, Waltham, USA) at 37 °C and 5% CO_2_ [48].

RAW264.7 cells were cultured in 96-well plates at a 5 × 10^5^ cells/mL density and exposed to LPS for 24 h at 0, 0.1, 0.5, 1, 1.5, and 2 µg/mL. The CCK8 cell proliferation and cytotoxicity Assay Kit (Dojindo, Kumamoto, Japan) was used to determine the cell viability of FVP and FFVP (0, 25, 50, 75, 100, 125, and 150 µg/mL). Optical density values were measured at 450 nm using Microplate Reader.

#### 4.5.2. NO and ROS Analysis

The cultured cells were divided into the following groups: CON, LPS (1 µg/mL LPS), FVP, and FFVP (1 µg/mL LPS plus 25, 50, and 100 µg/mL FVP/FFVP). The NO contents and ROS levels in the supernatants were determined after centrifugation at 1000 rpm for 5 min and incubation for 24 h by measuring the fluorescence intensity at 485 nm and 540 nm, according to the instructions. Then, the NO contents and ROS levels were calculated according to standard curve and formula (1), respectively [49]. The cell fluorescence images were analyzed using a Nikon fluorescence microscope (ECLIPSE NI, Tokyo, Japan).
(1)$$\mathrm{ROS}\;(\%)\;=\;\frac{{\mathrm{FI}}_{T}{\;-\;\mathrm{FI}}_{B}}{{\mathrm{FI}}_{C}{\;-\;\mathrm{FI}}_{B}}\;\times \;100\%$$
FI_T_: fluorescence intensity of treatment group; FI_B_: fluorescence intensity of blank group; FI_C_: fluorescence intensity of control group.

#### 4.5.3. Phagocytic Analysis

The FITC-Dextran (1 mg/mL) was added to the cell culture medium of each group and cultured at 37 °C for 1 h [50]. Cells of each group were washed twice with ice-cold PBS and centrifuged at 1000 rpm for 5 min. The average fluorescence intensity of cells was determined using flow cytometry (BD LSRFORTESSA, Indianapolis, USA).

#### 4.5.4. Cytokines Analysis

The supernatants of the culture medium were collected and centrifuged at 3000 rpm for 10 min. The supernatants were then used for the determination of IL-1β, IL-6, IL-18, and TNF-α with the corresponding mice ELISA kit (MultiSciences, Shanghai, China) [51]. All samples were measured using known amounts of recombinant protein.

#### 4.5.5. Quantitative Real-Time PCR Analysis

Total RNA of RAW264.7 cells was isolated with the Trizol Reagent (Takara, Kyoto, Japan) after incubation for 24 h [52]. The spectrophotometer (GeneQuant 1300 GE, Austin, USA) determined the RNA amount and quality. The absorbance of samples between 1.8 and 2.0 at 260–280 nm exhibited acceptable quality and integrity. The cDNA was synthesized with TransScript First-Strand complementary DNA synthesis superMix (AT341–01, Roche, Basel, Switzerland). The primer sequences are shown in Table 1. PCR was performed in a StepOnePlus Real-time system (ABI, Carlsbad, USA) using SYBR Green PCR Core Reagents (Roche, Basel, Switzerland). The housekeeping gene β-actin was used as a reference gene for normalization. Data were examined according to the 2^–△△Ct^ method. Results were expressed as relative mRNA levels.

#### 4.5.6. Western Blot Analysis

The cells were divided into CON (working medium), LPS (1 µg/mL LPS), FVP, and FFVP (1 µg/mL LPS plus 25, 50, and 100 µg/mL FVP/FFVP) groups after culturing for 12 h [53]. The RAW264.7 cells were then washed once in PBS and lysed in superactive RIPA buffer (Beyotime, Shanghai, China) with protease and phosphatase inhibitor cocktails (Sigma Co, St. Louis, MO, USA) for 10 min on a rocker at 4 °C. Protein samples were examined using sodium dodecyl sulfate-polyacrylamide gel electrophoresis (SDS-PAGE) and moved into polyvinylidene fluoride (PVDF) membranes (0.45 µm, Millipore, Billerica, USA). Each PVDF membrane was first blocked with Tris-buffered saline Tween (TBST) (100 mM Tris-HCl,150 mM NaCl, 0.05% Tween 20, pH 7.5) with 5% non-fat dried milk for 2 h, and then incubated with the following primary antibodies: p-p65 (1:800), p65 (1:800), p-IkBα (1:1000, Cell Signaling, Boston, MA, USA); IkBα (1:400, Wanleibio, Shenyang, China); NLRP3 (1:1500), Caspase-1 (1:800), IL-1β (1:400), and β-actin (1: 5000 SUNGENE biotech, Shanghai, China) overnight at 4 °C on a shaker (SC 390C, Shanghai Brave Construction Development co., Ltd, Shanghai, China). The goat anti-rabbit (1:3000, ZSGB-BIO, Beijing, China) horseradish peroxidase (HRP) conjugated secondary antibody was added and spotted using enhanced electrochemical luminescence (ECL) reagent (Beyotime, Shanghai, China). The β-actin was used as a protein loading control. Amersham Imager 600 (Cytiva, Montana, USA) was used for quantitative analysis.

### 4.6. Statistical Analysis

All analyses were carried out with the SPSS 22.0 software (SPSS Incorporated, Armonk, USA), and the statistical significance was examined using ANOVA followed by an LSD test. Data were expressed as mean ± standard deviation, and all measurements were conducted in triplicate. *p* < 0.05 and *p* < 0.01 were considered important and markedly significant in different groups.

## 5. Conclusions

Antioxidant and anti-inflammatory capacities of FVP and FFVP were first reported in this study. A new syringic acid, accounting for 22.22%, was produced in FFVP. Overall, the antioxidant capacity of FFVP was greater than that of FVP. Further, FFVP has higher anti-inflammatory abilities to enhance cell viability and phagocytosis, constrain the secretion of NO and ROS, and reduce the inflammatory response of RAW264.7 cells when compared to FVP. The possible mechanism of FFVP comprised a reduction in the activation of the NLRP3 inflammasome by reducing ROS secretion and preventing the activation of the NF-κB signaling pathway. This study indicated that FFVP could be used as a new polyphenols source, and be widely applied as antioxidant and anti-inflammatory agents in the food and health industry.

## Figures and Tables

**Figure 1 molecules-26-06205-f001:**
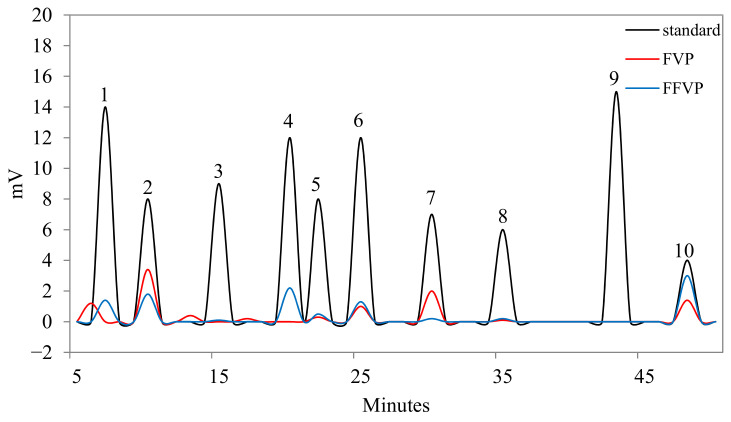
HPLC profiles of standard compounds, FVP and FFVP. (1) gallic acid; (2) chlorogenic acid; (3) caffeic acid; (4) syringic acid; (5) epicatechin; (6) ferulic acid; (7) rutin; (8) phloridzin; (9) resveratrol; (10) quercetin. HPLC: high performance liquid chromatography; FVP: *Flammulina velutipes* polyphenols; FFVP: fermented *Flammulina velutipes* polyphenols.

**Figure 2 molecules-26-06205-f002:**
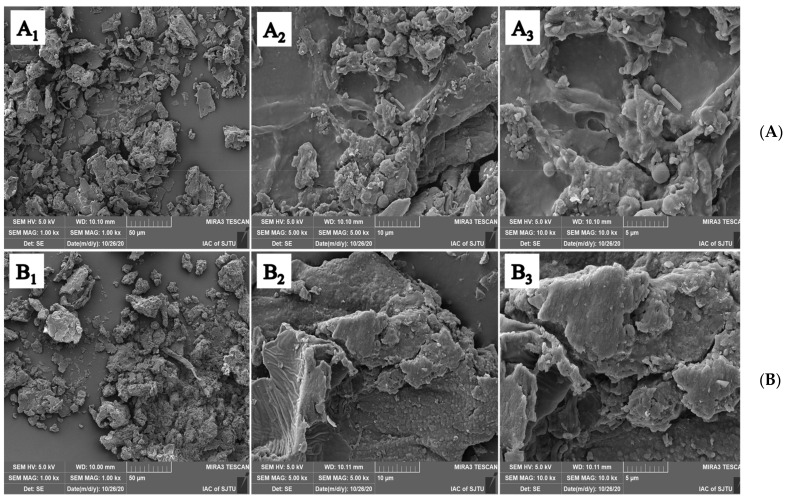
SEM images of FVP (**A**) and FFVP (**B**). A_1_/B_1_, A_2_/B_2_ and A_3_/B_3_ are magnifications of 1000×, 5000× and 10,000×, respectively. SEM: scanning electron microscope; FVP: *Flammulina velutipes* polyphenols; FFVP: fermented *Flammulina velutipes* polyphenols.

**Figure 3 molecules-26-06205-f003:**
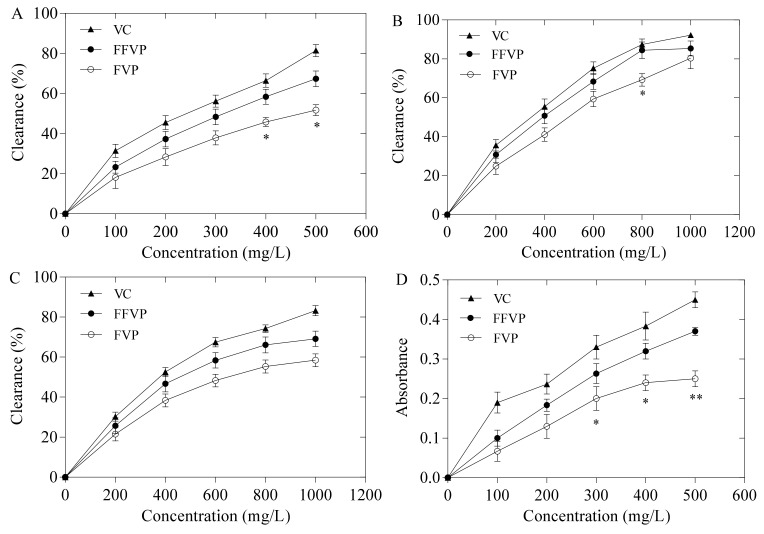
DPPH radical scavenging activity (**A**), OH· scavenging activity (**B**), O^2−^· scavenging activity (**C**) and total reducing power (**D**) of FVP and FFVP. FVP and FFVP are *Flammulina velutipes* polyphenols and fermented *Flammulina velutipes* polyphenols. * *p* < 0.05 and ** *p* < 0.01 indicate that antioxidant activity of FFVP is significantly and extremely significantly higher than that of FVP, respectively.

**Figure 4 molecules-26-06205-f004:**
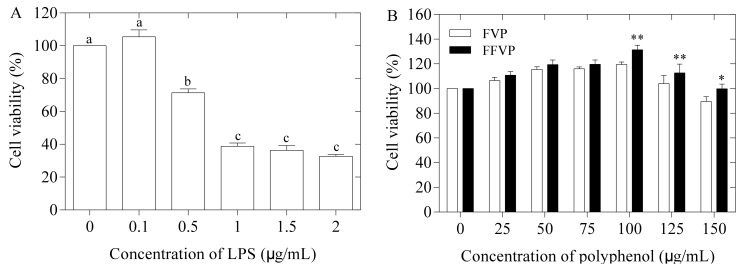
Effect of LPS (**A**), FVP and FFVP (**B**) on RAW264.7’s cell viability. FVP and FFVP are *Flammulina velutipes* polyphenols and fermented *Flammulina velutipes* polyphenols. a, b and c represent significant difference among cell viability when treated by different concentrations of LPS. * *p* < 0.05 and ** *p* < 0.01 indicate FFVP versus FVP treated cells.

**Figure 5 molecules-26-06205-f005:**
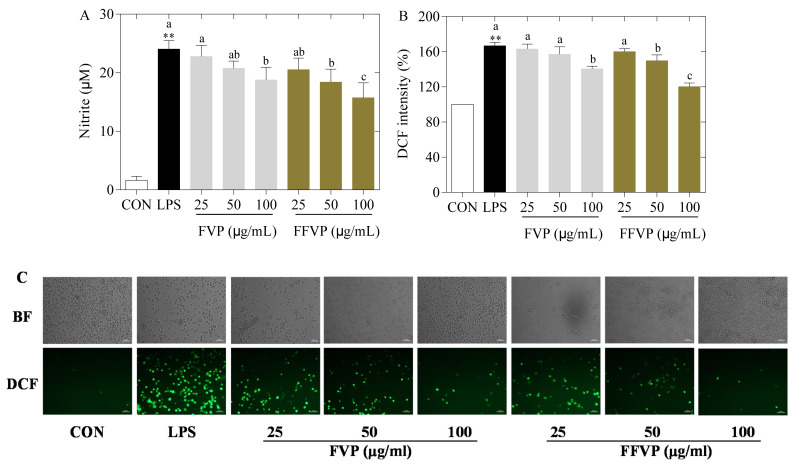
LPS-induced NO (**A**), DCF intensity (**B**) and fluorescence morphology (**C**) in RAW264.7 cells treated by FVP and FFVP. RAW264.7 cells were stimulated with 1 μg/mL LPS and/or 25, 50 and 100 μg/mL FVP/FFVP for 24 h. CON group, LPS group (1 μg/mL LPS), FVP and FFVP groups (1 μg/mL LPS plus 25, 50 and 100 μg/mL). BF: bright field; DCF: 2′,7′-Dichlorofluorescein. ** *p* < 0.01 represent LPS versus CON group; a–c represent FVP and FVP groups versus LPS-treated cells.

**Figure 6 molecules-26-06205-f006:**
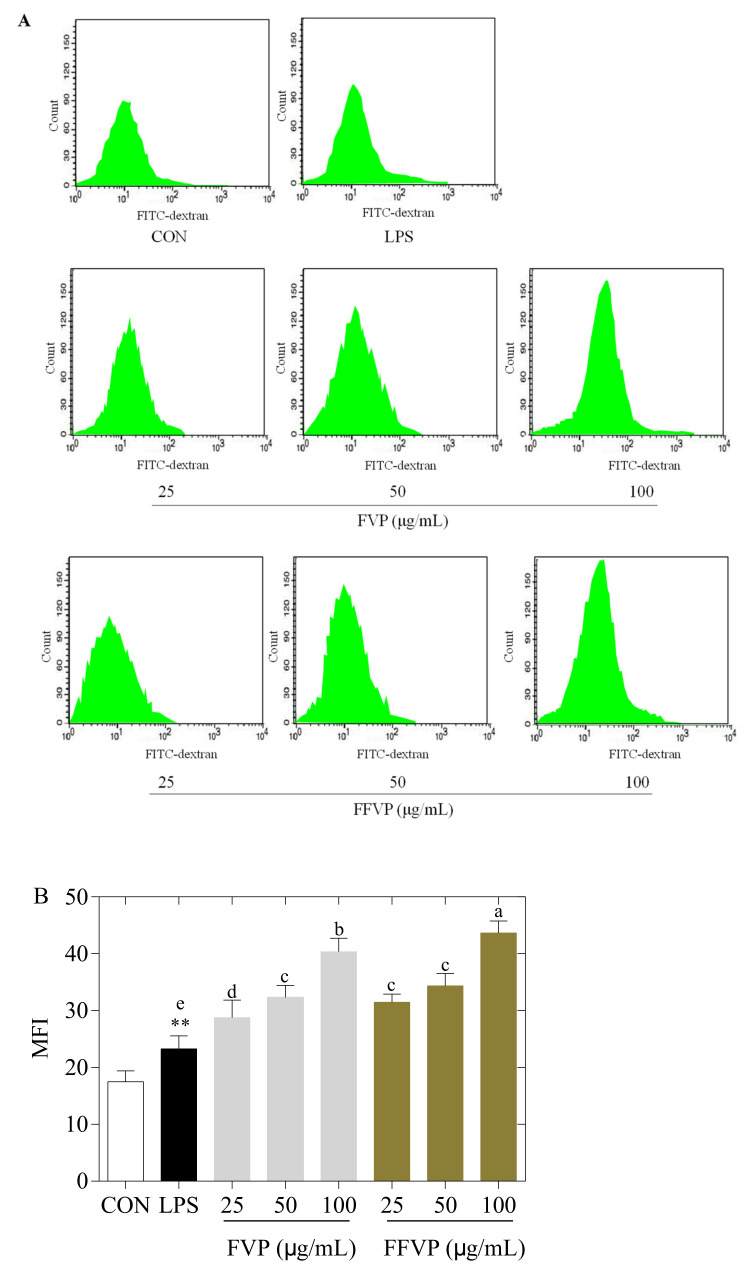
Flow cytometry analysis (**A**) and phagocytosis (**B**) of LPS-induced RAW264.7 cells treated by FVP and FFVP. RAW264.7 cells were stimulated with 1 μg/mL LPS and/or 25, 50 and 100 μg/mL FVP/FFVP for 24 h. CON group, LPS group (1 μg/mL LPS), FVP and FFVP groups (1 μg/mL LPS plus 25, 50 and 100 μg/mL FVP/FFVP). MFI: mean fluorescence intensity; FVP and FFVP are *Flammulina velutipes* polyphenols and fermented *Flammulina velutipes* polyphenols. ** *p* < 0.01 represent LPS versus CON group; a–e represent FVP and FVP groups versus LPS-treated cells.

**Figure 7 molecules-26-06205-f007:**
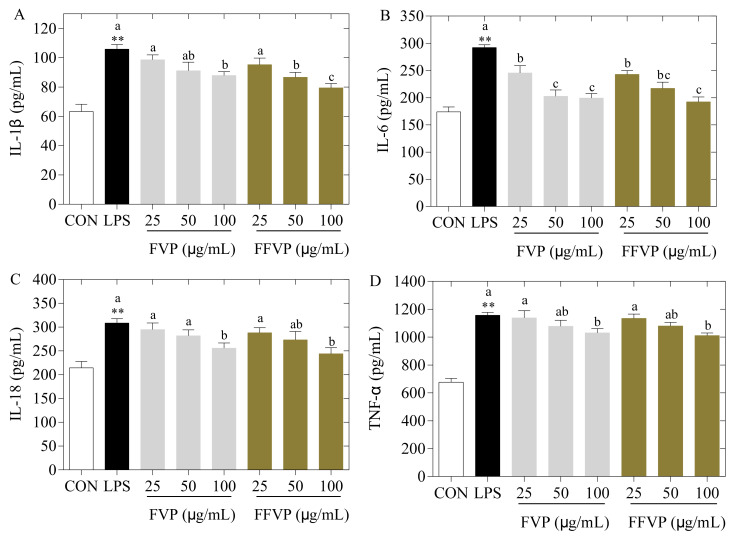
LPS-induced IL-1β (**A**), IL-6, (**B**), IL-18 (**C**) and TNF-α (**D**) secretion treated by FVP and FFVP. RAW264.7 cells were stimulated with 1 μg/mL LPS and/or 25, 50 and 100 μg/mL FVP/FFVP for 24 h. CON group, LPS group (1 μg/mL LPS), FVP and FFVP groups (1 μg/mL LPS plus 25, 50 and100 μg/mL FVP/FFVP). ** *p* < 0.01 represent LPS versus CON group; a–c represent FVP and FVP groups versus LPS-treated cells.

**Figure 8 molecules-26-06205-f008:**
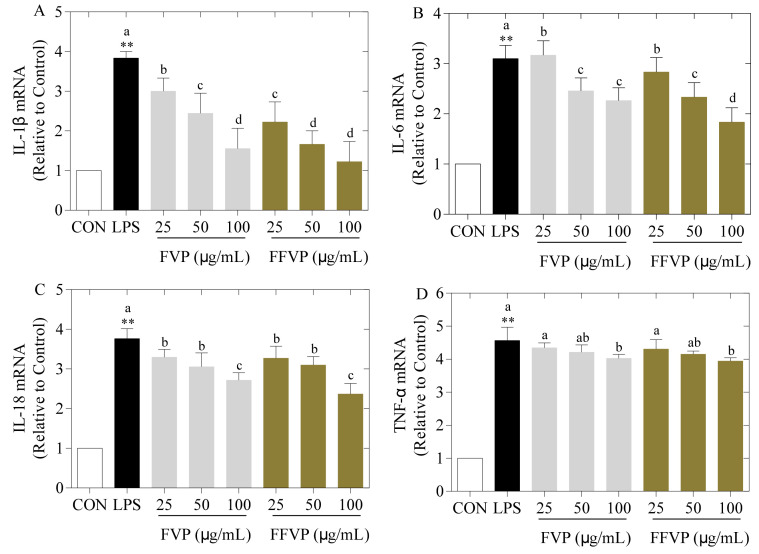
LPS-induced IL-1β (**A**), IL-6, (**B**), IL-18 (**C**) and TNF-α (**D**) expression treated by FVP and FFVP. RAW264.7 cells were stimulated with 1 μg/mL LPS and/or 25, 50 and 100 μg/mL FVP/FFVP for 24 h. CON group, LPS group (1 μg/mL LPS), FVP and FFVP groups (1 μg/mL LPS plus 25, 50 and 100 μg/mL FVP/FFVP). ** *p* < 0.01 represent LPS versus CON group; a–c represent FVP and FVP groups versus LPS-treated cells.

**Figure 9 molecules-26-06205-f009:**
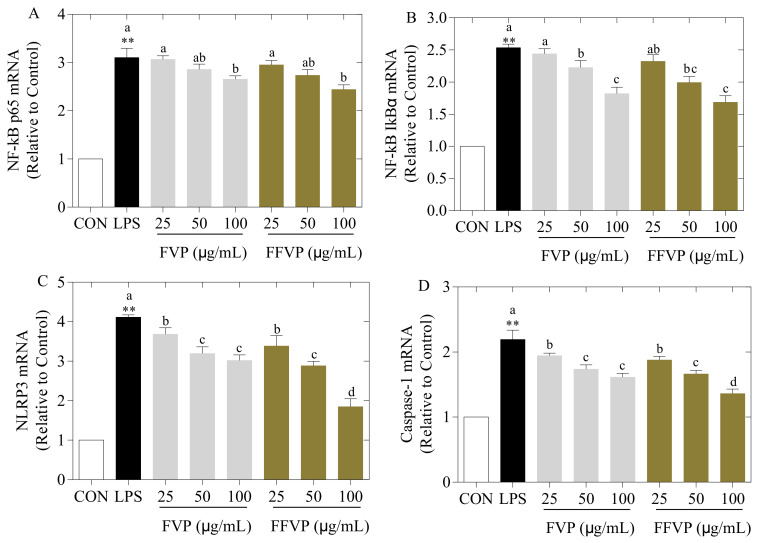
LPS-induced NF-κB p65 mRNA (**A**), NF-κB IkBα mRNA (**B**), NLRP3 mRNA (**C**) and Caspase-1 mRNA (**D**) expression treated by FVP and FFVP. RAW264.7 cells were stimulated with 1 μg/mL LPS and/or 25, 50 and 100 μg/mL FVP/FFVP for 24 h. CON group, LPS group (1 μg/mL LPS), FVP and FFVP groups (1 μg/mL LPS plus 25, 50 and 100 μg/mL FVP/FFVP). ** *p* < 0.01 represent LPS versus CON group; a–c represent FVP and FVP groups versus LPS-treated cells.

**Figure 10 molecules-26-06205-f010:**
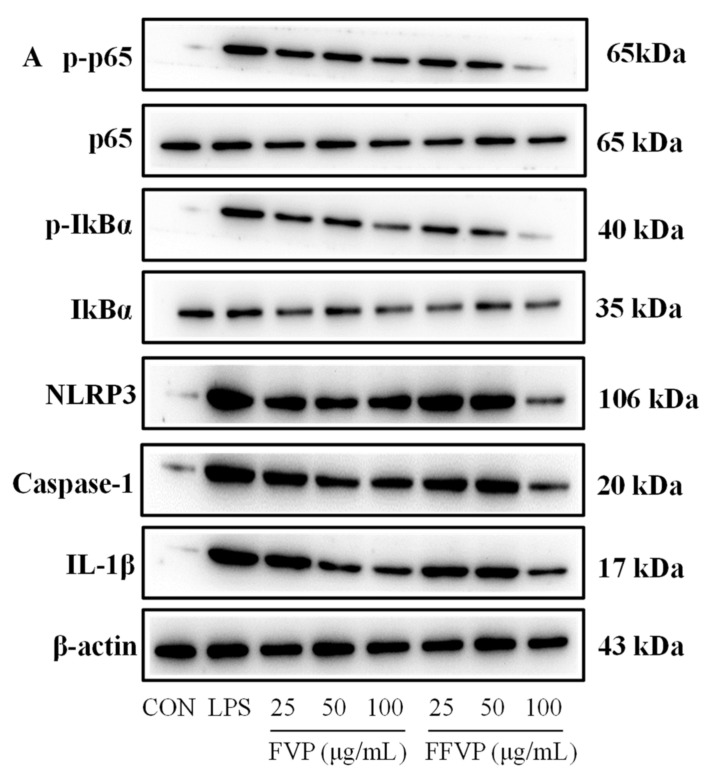
LPS-induced NF-κB p-p65/p65 (**A**,**B**), p-IkBα/IkBα (**A**,**C**), NLRP3/β-actin (**A**,**D**), Caspase-1/β-actin (**A**,**E**) and IL-1β/β-actin (**A**,**F**) protein expression treated by FVP and FFVP. RAW264.7 cells were stimulated with 1 μg/mL LPS and/or 25, 50 and 100 μg/mL FVP/FFVP for 24 h. CON group, LPS group (1 μg/mL LPS), FVP and FFVP groups (1 μg/mL LPS plus 25, 50 and 100 μg/mL FVP/FFVP). ** *p* < 0.01 represent LPS versus CON group; a–e represent FVP and FVP groups versus LPS-treated cells.

**Table 1 molecules-26-06205-t001:** Content of phenolic compounds in FVP and FFVP.

Phenolic Compound	FVP	FFVP
Gallic acid (%)	13.33 ± 0.87 ^a,b^	14.14 ± 1.20 ^b^
Chlorogenic acid (%)	37.78 ± 2.89 ^d^	18.18 ± 1.11 ^c^
Syringic acid (%)	0.00	22.22 ± 1.94 ^d^
Ferulic acid (%)	11.11 ± 0.76 ^a^	13.13 ± 0.95 ^b^
Rutin (%)	22.22 ± 1.75 ^c^	2.02 ± 0.13 ^a^
Quercetin (%)	15.56 ± 1.06 ^b^	30.30 ± 2.17 ^e^

In the same column, values with same superscript letter (^a–e^) were not significantly different (*p* > 0.05), values with different superscript letter were significantly different (*p* < 0.05).

## Data Availability

Data are contained within the article.

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
