# Peer review of "Characterization, Antioxidant and Anti-Inflammation Capacities of Fermented Flammulina velutipes Polyphenols"

_molecules, 2021, doi:10.3390/molecules26206205_

Round 1

Reviewer 1 Report

Dear Authors,

I read the manuscript " Characterization, antioxidant and anti-inflammation capacities of fermented Flammulina velutipes polyphenols” by Sheng Ma, Hongcai Zhang and Jianxiong Xu.

The article is generally clear, the experiments and analyses are well conducted. I would suggest to put the “results and discussion” after the “materials and methods” chapter to ease the reading.

Other question or mistake are here reported:

line 11: you wrote about new eugenic acid. It is syringic acid you found in FFVP, isn’t it?

line 114: the reference (Valerio et al., 2017) should be erase

line 119-121: It is not clear how the variation of absorbance at 3450 and 1041 could be related with the cell membrane degradation.  Generally, in literature the variation of 2850 nm peak is considered more relevant to confirm the cell membrane degradation. 

line 122: I would suggest authors to start the phrase stressing that you are talking about UV spectra from now on.

line 133: You analyze DPPH antioxidant capacity using concentration between 400 and 500 mg/L. Also in other experiments the concentration of the polyphenols used were very high. Normally, the blood concentration of antioxidant are around 10-30 mg/L and experiments on cell use a range between 10 and 100 mg/L. Please clarify this choice.

line 180-185: you did not analyze the enhancement of phagocytosis in treated RAW264.7 cells

line 209: year in brackets should be cancelled.

line 261: year in brackets should be cancelled

line 271: year in brackets should be cancelled

line 349-350: in the experimental part you refer to mgGAE/g as the method of measuring the antioxidant capacity in DPPH analyses, but in paragraph 2.4 and in Figure 4 DPPH data are presented in another way. Please clarify this inconsistency.  

line 392-393: FIT and FIC are written without subscripts.

line 448: again you wrote about new eugenic acid instead of syringic acid

line 452: “This study indicated that FFVP as new polyphenols” This phrase is wrong. Do you mean that FFVP is a new polyphenols source?

line 446-453: the conclusion are very short and should be deepen.

Author Response

Review 1

Dear Authors,

I read the manuscript " Characterization, antioxidant and anti-inflammation capacities of fermented Flammulina velutipes polyphenols” by Sheng Ma, Hongcai Zhang and Jianxiong Xu.

The article is generally clear, the experiments and analyses are well conducted. I would suggest to put the “results and discussion” after the “materials and methods” chapter to ease the reading.

Thanks for your suggestion. However, the journal “Molecules” requires the “results and discussion” section to precede the “materials and methods” section.

Other question or mistake are here reported:

line 11: you wrote about new eugenic acid. It is syringic acid you found in FFVP, isn’t it?

Yes, it is syringic acid. We’ve changed it as “syringic acid”.

line 114: the reference (Valerio et al., 2017) should be erase

We’ve erased the reference (Valerio et al., 2017).

line 119-121: It is not clear how the variation of absorbance at 3450 and 1041 could be related with the cell membrane degradation. Generally, in literature the variation of 2850 nm peak is considered more relevant to confirm the cell membrane degradation.

We’ve changed “3450 and 1041” to “2850”.

line 122: I would suggest authors to start the phrase stressing that you are talking about UV spectra from now on.

We added this sentence “The UV spectrum of FVP and FFVP are shown in Fig. 3B”.

line 133: You analyze DPPH antioxidant capacity using concentration between 400 and 500 mg/L. Also in other experiments the concentration of the polyphenols used were very high. Normally, the blood concentration of antioxidant are around 10-30 mg/L and experiments on cell use a range between 10 and 100 mg/L. Please clarify this choice.

In order to study the antioxidant activities of FVP and FFVP, we first conducted in vitro experiments, and found that there was no significant difference between FVP and FFVP at low concentration, but significant difference after increasing the concentration. Then cell tests were carried out and significant differences were found between FVP and FFVP at low concentrations.

line 180-185: you did not analyze the enhancement of phagocytosis in treated RAW264.7 cells

We cited Cristina’s study to analyze the enhancement of phagocytosis in treated RAW264.7 cells.

line 209: year in brackets should be cancelled.

We’ve cancelled the “year” in brackets.

line 261: year in brackets should be cancelled

We’ve cancelled the “year” in brackets.

line 271: year in brackets should be cancelled

We’ve cancelled the “year” in brackets.

line 349-350: in the experimental part you refer to mgGAE/g as the method of measuring the antioxidant capacity in DPPH analyses, but in paragraph 2.4 and in Figure 4 DPPH data are presented in another way. Please clarify this inconsistency. 

We’ve changed the wrong method, which is consistent with paragraph 2.4 and in Figure 4 DPPH data now.

line 392-393: FIT and FIC are written without subscripts.

FIT and FIC are written with subscripts now.

line 448: again you wrote about new eugenic acid instead of syringic acid

We’ve changed it as “syringic acid”.

line 452: “This study indicated that FFVP as new polyphenols” This phrase is wrong. Do you mean that FFVP is a new polyphenols source?

Yes, we’ve changed this sentence as “FFVP as new polyphenols source”.

line 446-453: the conclusion are very short and should be deepen.

We enriched the conclusion and make it more deepen.

Reviewer 2 Report

In this study, the authors investigated the effects of FVP and fermented FVP (FFVP) in antioxidant and anti-inflammation activities. It’s an interesting study, however, some questions need to be considered.

Comments      

  1. Line 14 to line 15, the method to describe the mRNA expression is rarely seen. No need to use specific data in the abstract.
  2. Combine fig.1A and B. and use different colors to get a clear comparison.
  3. There should be a positive control for antioxidant measurement and calculate the IC50 of FVP and FFVP.
  4. In fig.6C, the brightfield images are hard to see. Please use a white background.
  5. The discussion is weak. The discussion should focus on the FVP and FFVP and discuss the reason why fermented FVP is better than FVP, and how do they suppress oxidants and inflammation?

Author Response

Review 2

In this study, the authors investigated the effects of FVP and fermented FVP (FFVP) in antioxidant and anti-inflammation activities. It’s an interesting study, however, some questions need to be considered.

Line 14 to line 15, the method to describe the mRNA expression is rarely seen. No need to use specific data in the abstract.

We’ve deleted the specific data in the abstract.

Combine fig.1A and B. and use different colors to get a clear comparison.

The fig.1A and B has been combined.

There should be a positive control for antioxidant measurement.

We added a positive control for antioxidant measurement.

In fig.6C, the bright field images are hard to see. Please use a white background.

We used a white background for the bright field images.

The discussion is weak. The discussion should focus on the FVP and FFVP and discuss the reason why fermented FVP is better than FVP, and how do they suppress oxidants and inflammation?

We’ve discuss the reason why fermented FVP is better than FVP.

Reviewer 3 Report

This paper well shows the good properties (antioxidant, anti-inflammation capacities, etc.) of fermented FVP (FFVP). 
It is thought that these characteristics of FFVP were rarely reported.

However, I think that only some notation needs to be corrected.

Tables 1 and 2 show the contents of gallic acid, chlorogenic acid, syringic acid, ferulic acid, rutin, and quercetin, which are components of FVP and FFVP.
Therefore, it is considered more appropriate to indicate the units indicated next to FVP and FFVP (% in Table 1, g/mol in Table 2) next to each phenol compound.

Author Response

Review 3

This paper well shows the good properties (antioxidant, anti-inflammation capacities, etc.) of fermented FVP (FFVP). It is thought that these characteristics of FFVP were rarely reported. However, I think that only some notation needs to be corrected.

Tables 1 and 2 show the contents of gallic acid, chlorogenic acid, syringic acid, ferulic acid, rutin, and quercetin, which are components of FVP and FFVP. Therefore, it is considered more appropriate to indicate the units indicated next to FVP and FFVP (% in Table 1, g/mol in Table 2) next to each phenol compound.

Thanks for your suggestion, the units has indicated next to each phenol compound.

Round 2

Reviewer 2 Report

Thanks for the responses.

Author Response

Thanks for your comments and suggestions.